# Bayesian Reinforcement Learning for Single-Episode Missions in Partially Unknown Environments

**Matthew Budd    Paul Duckworth    Nick Hawes    Bruno Lacerda**
Oxford Robotics Institute, University of Oxford
{mbudd, pduckworth, nickh, bruno}@robots.ox.ac.uk

**Abstract:** We consider planning for mobile robots conducting missions in real-world domains where *a priori* unknown dynamics affect the robot's costs and transitions. We study single-episode missions where it is crucial that the robot appropriately trades off exploration and exploitation, such that the learning of the environment dynamics is *just enough* to effectively complete the mission. Thus, we propose modelling unknown dynamics using Gaussian processes, which provide a principled Bayesian framework for incorporating online observations made by the robot, and using them to predict the dynamics in unexplored areas. We then formulate the problem of mission planning in Markov decision processes under Gaussian process predictions as Bayesian model-based reinforcement learning. This allows us to employ solution techniques that plan more efficiently than previous Gaussian process planning methods are able to. We empirically evaluate the benefits of our formulation in an underwater autonomous vehicle navigation task and robot mission planning in a realistic simulation of a nuclear environment.

**Keywords:** Planning under Uncertainty, Gaussian Processes, Single-Episode Bayesian Reinforcement Learning

## 1   Introduction

Real-world mobile robots rarely have complete knowledge of their environment dynamics. When operating under uncertainty, they need to be able to incorporate their online *observations* of uncertain environment features into their plans. In this paper, we consider the single-episode setting where a robot must carry out a mission in an environment for which the dynamics are not fully known at deployment time. The mission is specified by a goal state(s) that the agent must eventually reach while minimising incurred cost under an environment feature that has *a priori unknown dynamics*. The robot's information gathering capabilities are limited, as the environment features are only observable at the robot's current state. An example would be a Geiger counter-equipped robot minimising its cumulative radiation exposure in an environment with an unknown radiation distribution. In this setting, it is infeasible to pre-plan for every possible environment that might be encountered. Moreover, even if it were possible to pre-train a standard reinforcement learning (RL) agent on every possible environment, it would lack the flexibility to *adapt* the robot's behaviour taking into account new observations it can collect during the mission.

To model continuous environment features, we follow previous works [1, 2, 3, 4] and use a Gaussian process (GP) [5] to predict unknown dynamics away from the agent's current location. GPs are well-suited for modelling spatio-temporal distributions by incorporating online measurements into the posterior distribution, along with a measure of predictive uncertainty. GP prior hyperparameters can be estimated from physical intuition or from a small dataset of similar environments. Similarly to [1, 2, 3, 4], our GP maintains a *belief* over the underlying dynamics of the environment. Rather than ensuring safe environment exploration or maximising information collected, we extend these works by formulating a unified Bayes-optimal framework for efficient online mission planning.

We therefore pose our task as online, model-based Bayesian RL (BRL) with a GP belief over the transition function. As all environmental uncertainty is then encapsulated within the transition function, we assume that the cost function is known given an instance of the environment dynamics. Continuing the previous example, the robot does not need to learn online that it incurs more damage

6th Conference on Robot Learning (CoRL 2022), Auckland, New Zealand.

from higher levels of radiation. Our problem statement also assumes a discrete state space (except for unknown environment dynamics, which are continuous) and full observability of the current state.

The BRL formulation provides several advantages. First, it is known to optimally trade off exploration and exploitation [6], which is crucial in the single-mission setting we address: over-exploration may hinder mission performance or increase the risk of failure. Second, it allows us to intuitively encode the agent's transition function and local observability limitations, and reflects the existence of fixed (but *a priori* unknown) true underlying environment dynamics; third, it enables the use of efficient Monte-Carlo planning approaches developed for BRL.

Our contributions are to 1. formulate goal-driven planning in unknown environments as model-based BRL; 2. adapt the Bayes-adaptive Monte-Carlo planning (BAMCP) algorithm [6] for our GP-based goal-driven BRL problem formulation; and 3. use this formulation and algorithm to improve on previous GP planning approaches, both in expressibility and computational efficiency. In particular, we exploit techniques that allow us to efficiently sample possible environment dynamics from the GP to use during planning. To the best of our knowledge, we are the first to apply BRL with GP belief models to goal-driven planning in partially unknown environments.

## 2   Related Work

Non-myopic decision-making with an unknown transition function requires reasoning over possible observation (i.e. function evaluation) sequences. This task has been investigated from the perspective of several fields, including sequential Bayesian optimisation (BO) of unknown functions. An example BO objective could be to stay within a computational evaluation budget while improving a GP model of the unknown function. Some recent methods are able to pose multiple-step look-ahead in GPs as a single joint optimisation problem [7]. However, we focus on physical systems such as mobile robots which are required to physically move and observe any queried location: observations cannot be made in parallel or at freely specified locations. This implies reachability, observability and cost limitations that are not usually considered when using BO.

Considering some of these aspects, recent literature [8, 3] performs non-myopic "informative path planning" for environmental monitoring. Similar to our approach, these methods perform Monte-Carlo tree search (MCTS) [9] in belief space with a GP belief. However, they assume robot actions have deterministic outcomes, and do not consider the case where the unknown environment features can affect robot transition dynamics, as we do. This recent literature uses a partially observable Markov decision process (POMDP) [10] problem formulation, and plan with MCTS trees of computationally expensive GP-represented beliefs due to their BO objective. Our Bayes-adaptive MDP (BAMDP) [11] formulation more appropriately represents the existence of fixed but *a priori* unknown environment dynamics. In the context of a goal-based planning objective, the two formulations and solution methods are equivalent, as we demonstrate in this paper. However, we use model-based BRL techniques to *root sample* the GP environment belief. This avoids computationally expensive belief updates, enabling the construction of larger MCTS trees within the same computational budget.

Monte-Carlo tree search in GP-modelled unknown environments has also been carried out in [12], where the environment features affect only transition durations in a semi-MDP. A model-based BRL method with a Gaussian process dynamical model (GPDM) belief representation was presented in [13]. Their method must make restrictive maximum likelihood transition/observation assumptions for tractability, due to their unfocused Monte-Carlo planning algorithm. Alternatively, some RL techniques such as Gaussian Process temporal difference [14] use GPs to directly model an MDP value function. We argue that GP modelling the real-world environmental phenomena, rather than the value function, lets us provide physically principled and interpretable prior knowledge.

## 3   Preliminaries

**Markov Decision Processes.**   We consider stochastic shortest path (SSP) MDP problems [15], as they are well-suited for specifying single episode goal-driven missions. An SSP MDP is defined as a tuple $\mathcal{M} = \langle S, s_0, A, T, C, G \rangle$, where $S$ is a finite set of states; $s_0 \in S$ is the initial state; $A$ is a finite set of actions; $T : S \times A \times S \to [0, 1]$ is a probabilistic transition function $T(s, a, s') = p(s' \mid s, a)$; $C : S \times A \to \mathbb{R}_{\geq 0}$ is a cost function; and $G \subset S$ is a set of absorbing, zero-cost goal states. A history $h$ of an MDP $\mathcal{M}$ is a state-action sequence $s_0 a_0 s_1 a_1 \cdots a_{t-1} s_t$ such that $T(s_i, a_i, s_{i+1}) > 0$

for all $i \in \{0, \cdots, t-1\}$. We denote the set of all histories of $\mathcal{M}$ as $\mathcal{H}^{\mathcal{M}}$. A stationary, deterministic policy is a mapping $\pi : S \rightarrow A$ that defines the action to take at each state. A policy is proper in a state $s$ if it reaches a goal state $s_g \in G$ when starting from $s$ with probability 1. In an SSP MDP there must exist a policy that is *proper* in all states, and all improper policies must incur infinite cost. Under these assumptions, there exists a cost-optimal proper policy [16].

**Bayesian RL.** In BRL, an agent uses Bayesian inference to maintain a posterior distribution, or *belief*, over the true dynamics of the underlying model given some prior distribution. For an MDP, either or both of the transition function $T$ and cost function $C$ could be a priori unknown. We focus on the case where only $T$ is unknown. $C$ can be assumed to be known in our setting, as it represents the known effect of a *given instance* of environment dynamics on the robot. For example, the time cost of travelling against given water current vectors can be calculated given the value of the vectors and the vehicle's known dynamics. Given a history $h = s_0 a_0 s_1 a_1 \cdots a_{t-1} s_t$, it is possible to generate the posterior belief over the transition function $T$ given $h$. This can be carried out with successive applications of Bayes' rule $p(T \mid h_i) \propto p(h_i \mid T) p(T)$ from the initial history $h_0 = s_0$ up to the full history $h_t = h$. This allows for the definition of a Bayes-adaptive MDP (BAMDP) [11], which achieves Bayes optimality by adding histories to its state representation, and encoding uncertainty over $T$ in its transition function. Let $\mathcal{M} = \langle S, s_0, A, T, C, G \rangle$ be an MDP with a prior belief $p(T)$ over the true transition function $T$. The corresponding BAMDP is an MDP $\mathcal{M}^+ = \langle S^+, s_0^+, A, T^+, C^+, G^+ \rangle$, where $S^+ = S \times \mathcal{H}^{\mathcal{M}}$; $s_0^+ = (s_0, h_0)$; $C^+((s, h), a) = C(s, a)$; $G^+ = \{(s, h) \in S^+ \mid s \in G\}$; and

$$T^+((s, h), a, (s', has')) = \int_T T(s, a, s') p(T \mid h) dT. \tag{1}$$

Although the state-history pairs in $S^+$ are redundant because the current state can be extracted from the history, we use the $(s, h)$ notation for clarity as in [6]. A policy in a BAMDP is a mapping $\pi : S \times \mathcal{H}^{\mathcal{M}} \rightarrow A$. The optimal policy $\pi^*$ minimises the expected cumulative cost to reach $G^+$, given the prior over $T$. This policy is stationary in $S^+$ but is history-dependent in the original MDP. $\pi^*$ considers the posterior $p(T \mid h)$ and adapts its action selection to account for the conditional distribution of $T$ given the observed $h$.

**Gaussian Processes.** A GP is a collection of random variables, any finite number of which have a joint Gaussian distribution [5]. A GP regression is of the form $o(s) \sim \mathcal{GP}(m(s), k(s, s'))$, giving a probability distribution over functions fully specified by the mean $m(s)$ and kernel $k(s, s')$ functions. We can let $m(s) = 0$ without loss of generality. Given a dataset of $n$ noisy observations $\mathcal{D} = \{(s_i, o(s_i) + \epsilon_i)\}_{i=1}^n$ for locations $s_i$ and where $\epsilon_i \sim \mathcal{N}(0, \sigma_\eta^2)$ is Gaussian observation noise, GP regression predicts unknown environment feature values at all inputs $s_*$. The kernel function $k$ is parameterised by hyperparameters $\boldsymbol{\theta}$. Given hyperparameter priors $p_0(\boldsymbol{\theta})$, their values are commonly optimised by maximising the log marginal likelihood for the model given the dataset. The resulting Gaussian posterior, conditioned on the observations $\mathbf{o} = [o(s_1) + \epsilon_1, \ldots, o(s_n) + \epsilon_n]^{\mathsf{T}}$, is a multivariate normal $p^{\mathcal{GP}}(o(s_*) \mid s_*, \mathcal{D})) \sim \mathcal{N}(\boldsymbol{\mu}_*, \boldsymbol{\Sigma}_*)$, where $\boldsymbol{\mu}_* = \mathbf{K}_*^{\mathsf{T}}(\mathbf{K}_n + \sigma_\eta^2 \mathbf{I})^{-1} \mathbf{o}$, and $\boldsymbol{\Sigma}_* = \mathbf{K}_{**} - \mathbf{K}_*^{\mathsf{T}}(\mathbf{K}_n + \sigma_\eta^2 \mathbf{I})^{-1} \mathbf{K}_*$.

The positive semi-definite kernel matrix $\mathbf{K}_n = [k(s, s')]_{s, s' \in \mathbf{s}_n}$, $\mathbf{K}_* = [k(s, s')]_{s \in \mathbf{s}_n, s' \in \mathbf{s}_*}$, $\mathbf{K}_{**} = [k(s, s')]_{s, s' \in \mathbf{s}_*}$, and $\mathbf{I} \in \mathbb{R}^{n \times n}$ is the identity matrix. We can sample functions from the GP posterior at a finite set of $m$ points, incurring $O((n + m)^3)$ computational cost.

## 4   Approach

### 4.1   Problem Formulation

In order to clearly separate known system transition dynamics from the unknown environment dynamics, we represent the unknown environment and its effect on the agent as an *MDP with Unknown Feature Values (U-MDP)* [4].

Let $S_k$ be a set of state features with discrete, known values (e.g. pose of a robot in a grid map) and $S_e$ a set of state features with unknown values in $\mathbb{R}$ (e.g. the water current vector at a pose). Let $o : S_k \rightarrow S_e$ be an *a priori* unknown mapping that specifies the values $o(s_k) \in S_e$ observed at locations $s_k \in S_k$. An SSP U-MDP is a tuple $\mathcal{M}^o = \langle S^o, s_0^o, A^o, T^o, C^o, G^o \rangle$ where: $S^o = S_k \times S_e$;

$s_0^o$ is the initial state $s_0^o = (s_{k,0}, o(s_{k,0}))$; $A^o$ is a finite set of actions; $T^o$ is the U-MDP transition function $T^o : (S_k \times S_e) \times A \times S_k \to [0, 1]$. As the state of the U-MDP is uniquely defined by the value of the known state feature(s) $s_k \in S_k$, the transition function of the U-MDP only represents the change in the known state feature(s); $C^o : S^o \times A \to \mathbb{R}_{\geq 0}$ is the cost function; and $G^o \subset S_k$ is the set of goal states, defined only across known value state features as $o(s_g)$ is not known for all $s_g \in G^o$. The problem addressed in this paper is formalised as an SSP U-MDP. The objective is to find a policy that minimises the expected cost to a reach state $(s_k, o(s_k)) \in S^o$ such that $s_k \in G^o$.

## 4.2   From U-MDPs to GP-BAMDPs

For notational simplicity, in the following we assume a single state feature with unknown values. The approach presented below can easily be extended to cases with more than one unknown value state feature, either using a multi-output GP [17] or multiple single-output GPs. The former assumes non-independent feature dynamics, where learning about one could improve predictions of another.

To estimate the unknown mapping $o$, we propose that the agent maintains a GP model created by adding a new observation of $o$ at each timestep. Specifically, after observing history $h = (s_{k,0}, s_{e,0})a_0(s_{k,1}, s_{e,1})a_1...a_{t-1}(s_{k,t}, s_{e,t})$, we define $\mathcal{D}_h = \{(s_{k,i}, s_{e,i}) \mid i \in \{0, \ldots, t\}\}$. Then, the GP model is denoted as $\mathcal{GP}_{\mathcal{D}_h}$ and the GP posterior over $o(s_k)$ is given by $p^{\mathcal{GP}}(s_e \mid s_k, \mathcal{D}_h)$. We assume that observations of $o$ have negligible noise, which corresponds to full observability of the current state. Finally, note that by modelling unknown environment features with a GP we are implicitly making regularity and Lipschitz continuity modelling assumptions on the environment feature functions [5].

We now formulate SSP U-MDP as a BAMDP with GP belief over the transition function.

**Proposition 1.** *Let $\mathcal{M}^o = \langle S^o, s_0^o, A^o, T^o, C^o, G^o \rangle$ be a U-MDP. The GP-BAMDP for $\mathcal{M}^o$ is a BAMDP $\mathcal{M}^{o+} = \langle S^{o+}, s_0^{o+}, A^o, T^{o+}, C^{o+}, G^{o+} \rangle$ where the transition function incorporates the GP posterior. Formally, for $s = (s_k, s_e)$, $s' = (s'_k, s'_e) \in S^o$; $a \in A$; and $h = (s_{k,0}, s_{e,0})a_0(s_{k,1}, s_{e,1})a_1...a_{t-1}(s_{k,t}, s_{e,t}) \in \mathcal{H}^{\mathcal{M}^o}$ for $(s_{k,t}, s_{e,t}) = (s_k, s_e)$:*

$$T^{o+}((s, h), a, (s', has')) = T^o((s_k, s_e), a, s'_k) \cdot p^{\mathcal{GP}}(s'_e \mid s'_k, \mathcal{D}_h). \tag{2}$$

*Proof.* We start by noting that the integral in Equation (1) is the product of two components. The first component is a specific possible transition function. As the single unknown component of $T^o$ is the mapping $o$, the value of $T^o(s, a, s')$ given knowledge of $o$ is defined according to the U-MDP transition function, ensuring that the unknown state feature dynamics are consistent with $o$:

$$T^o((s_k, s_e), a, (s'_k, s'_e) \mid o) = T^o((s_k, s_e), a, s'_k) \mathbb{I}[s'_e = o(s'_k)], \tag{3}$$

where $\mathbb{I}[.]$ is the indicator function. The second component is the posterior distribution over possible transition functions given a history $h$, which in our case is the GP posterior $p^{\mathcal{GP}}$:

$$p(o(s'_k) \mid h) = p^{\mathcal{GP}}(o(s'_k) \mid s'_k, \mathcal{D}_h). \tag{4}$$

As the unknown state feature dynamics are deterministic given $o$, the integral over $T$ from Equation (1) only has value when the indicator function in Equation (3) is 1. This leads to Equation (2), where all uncertainty in $T$ is captured in the GP posterior over $o$, and $T^{o+}$ represents the combination of the U-MDP transition function and the GP belief over the values of the unknown state features. $\square$

## 4.3   Solving U-MDPs with BAMCP

**GP-BAMCP.**   Having framed the U-MDP mission planning problem as a BAMDP, we can exploit MCTS planning frameworks that were developed in this context. Specifically, we base our algorithm on BAMCP [6]. This algorithm plans in belief space[1], but builds search trees of action-observation histories rather than of belief states. The search tree consists of alternating state and action nodes and is constructed over the course of Monte-Carlo trials starting from the root node and sampling action outcomes from a generative model. BAMCP adapts the concept of *root sampling* from POMCP [18], where for each MCTS trial a transition function $T$ is sampled from the root belief node and used throughout the trial. Actions are chosen inside the search tree using a *tree policy*, most commonly UCT [9]. New leaf nodes' values are estimated heuristically by continuing the trial trajectory from the leaf node using a *rollout policy*.

---

[1]The distribution over the transition function $T$ in the case of BAMCP.

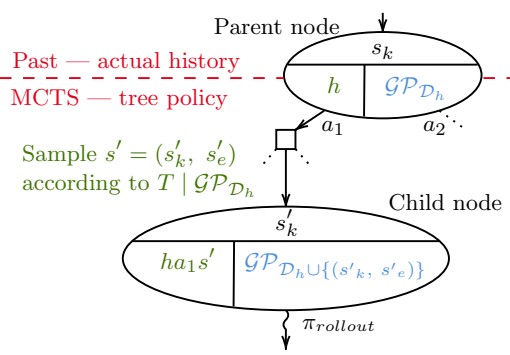

**Algorithm 1** GP-BAMCP: PLAN

1: **procedure** PLAN($\mathcal{GP}_{\mathcal{D}_h}$, history $h$, start state $(s_k, s_e)$, goal state set $G^o$)
2:    **repeat**
3:       $\hat{o} \leftarrow$ sample from $\mathcal{GP}_{\mathcal{D}_h}$
4:       $\hat{T}((s_k, s_e), a, (s'_k, s'_e))$       $\leftarrow$ $T^o((s_k, s_e), a, (s'_k, s'_e) \mid \hat{o})$
5:       SIMULATE($(s_k, s_e), h, \hat{T}, G^o$)
6:    **until** TIMEOUT( )
7:    **return** $\arg\min_a Q(h, a)$
8: **end procedure**

Figure 1: Example BAMDP MCTS search-tree, with search nodes as ellipses. Green shows search node generation/contents with a root sampling approach, and blue shows full belief planning.

Our online MCTS planning algorithm, a modified version of BAMCP for GP beliefs, is described in Algorithm 1. To simplify the presentation, we assume that the rollout policy is able to reach the goal set with probability 1. Our algorithm replaces BAMCP's depth $d$ and reward $R$ parameters with the goal set $G$ and costs $C$, respectively, to reflect the SSP mission setting of our work. In our case root sampling of $T$ is performed by sampling from the current GP posterior, as described in lines 3 and 4. Concretely, we take a sample $\hat{o} \sim \mathcal{GP}_{\mathcal{D}_h}$ where $\hat{o} : S_k \rightarrow S_e$ is a possible mapping from known states to values of the unknown state features. Sampling $\hat{o}$ from the root belief node corresponds to sampling a possible environment that is consistent with the current GP environment model, i.e. trained only on real-world observations. Finally, in line 7, the agent greedily selects a single real-world action by running MCTS trials up to a computational budget.

**Treatment of continuous $s_e$.** Equation (2) represents a BAMDP with both discrete (known value) and continuous (unknown value) state features, and therefore also a combination of a discrete transition function, $T^o$, and a continuous transition function given by the GP belief over $o$. This presents challenges for MCTS methods, since the probability of transitioning to the same state twice is 0. To enable generalisation between histories and allow the search tree to reach depths greater than 1, our algorithm aggregates similar outcomes in $s_e$ into the same child search node. Each search node has an associated "mean" value $\overline{s_e}$, where all outcomes $\|s'_e - \overline{s_e}\|_1 < \epsilon$ will be associated with that node. When two MCTS trials sample $s = (s_k, s_e)$ and $s' = (s_k, s'_e)$ from the same start state and action, if $\|s_e - \overline{s_e}\|_1 < \epsilon$ and $\|s'_e - \overline{s_e}\|_1 < \epsilon$, the two histories will be associated with the same child node. Note that this only associates histories to nodes and does not discretise the computation of reward or transition probabilities. The $\epsilon$ parameter therefore controls the search-tree branching factor, similarly to $\epsilon$ in [12] and the two branching factor parameters in progressive widening MCTS [19].

### 4.4 Theoretical Analysis

**Equivalence to Partially Observable MDP Belief Planning Methods.** Figure 1 depicts an example MCTS search-tree mid-mission. Several real actions have been taken in the environment, corresponding to the history $h$ in the root node. When carrying out BAMCP root sampling, a new leaf node is added to the MCTS tree by appending the parent node history with a new state. This new state $s \sim T(s, a, \cdot)$ is sampled from the MDP transition function which was itself sampled from the root for this MCTS simulation, in lines 3 and 4 in Algorithm 1. This is depicted in green in Figure 1.

Several previous approaches [8, 3] that integrate GPs and MDPs for decision-making in uncertain environments use a partially observable MDP (POMDP) [10]. We call these *POMDP belief MCTS* approaches. These methods incorporate a GP environment belief into the POMDP state, and use MCTS to plan in belief space. Here, new belief state leaf nodes are generated using a *belief update* from the parent node. Belief update in this context is carried out by sampling a hypothesised data point $\tilde{o} \sim p^{\mathcal{GP}}(\cdot \mid s_k, \mathcal{D}_{parent})$ from the parent node GP belief posterior and augmenting the child belief node's GP dataset with this new point: $\mathcal{D}_{parent} \cup \{(s_k, \tilde{o})\}$. This is shown in blue in Figure 1,

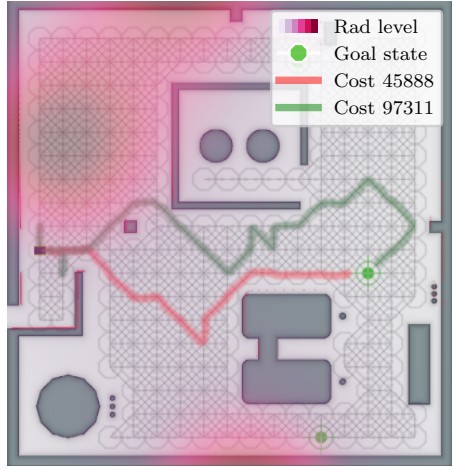 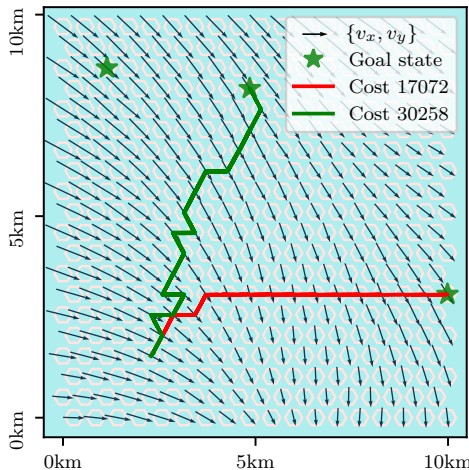

(a) Example radiation domain visualisation, with a robot trajectory generated by GP-SSP-BAMCP (red) and by GP belief MCTS (green) algorithms.

(b) Example ocean currents domain visualisation, with an AUV trajectory generated by GP-SSP-BAMCP (red) and by GP belief MCTS (green) algorithms.

Figure 2: Experiment domains. Randomly selected multi-goal problem instances are shown.

where the GP in the child node can be explicitly generated by a belief update from the parent node GP. GP belief updates require adding a single new data point to the model, the complexity of which can be reduced from $O(N^3)$ to $O(N^2)$ where $N$ is the number of data and sample points [17]. Even with this reduction, the belief update is still computationally expensive as the dataset of real and hypothesised observations grows. Furthermore, the belief updates are required sequentially, once per MCTS simulation to generate a new leaf node. This greatly slows the MCTS procedure.

In contrast, our method does not explicitly generate these hypothesised GPs and plans only using histories. The child node in Figure 1 still represents the same belief as the GP shown in blue, but only contains the history obtained using a root sampled transition function. We prove the history-GP equivalence, and hence the validity of root sampling in our setting, by showing that the probability of generating a history from the BAMDP is the same for root sampling as it is for maintaining and updating full GP belief at each belief node.

**Proposition 2.** *Let $\mathcal{P}_\pi^{h_t}(h_{t+\tau})$ be the probability of a history $h_{t+\tau}$ in the BAMDP, starting at history $h_t$ under policy $\pi$, when carrying out individual GP belief updates at every stage; and $\tilde{\mathcal{P}}_\pi^{h_t}(h_{t+\tau})$ be the probability of $h_{t+\tau}$ when carrying out GP root sampling. Then, $\mathcal{P}_\pi^{h_t}(h_{t+\tau}) = \tilde{\mathcal{P}}_\pi^{h_t}(h_{t+\tau})$ for all policies $\pi$ and all histories $h_{t+\tau}$.*

The proof is given in the appendix and is a direct adaptation of Lemma 1 in [20], accounting for the GP posterior over the transition function, and applies to a general stochastic policy $\pi : \mathcal{H}^\mathcal{M} \times A \to [0, 1]$.

Once any new real observations have been added to the root node GP, one can draw an arbitrary number of root samples to plan with at little additional computational cost. This means we can run more trials with the same computational budget, thus building a larger MCTS search tree, as we demonstrate in Section 5. Finally, we note that, due to the BO setting of [8, 3], where the objective is directly related to the uncertainty in the function being predicted by the GP, it is not enough to ensure the same distribution over histories. Thus, they must maintain full GP beliefs in their search nodes.

## 5 Experiments

**Domains Description.** We experimentally evaluate the proposed method in two simulated domains.

**1) Radiation domain**: a robot must navigate to a goal location (single-goal variant), or one of three goal locations (multi-goal variant), in a 20m × 20m reactor room with an unknown distribution of radiation level, while minimising its cumulative exposure. The GP model is log-warped [21] to constrain the predictions to be strictly positive and better model order-of-magnitude variation

in radiation level caused by $1/r^2$ "solid angle" radiation physics. Goal locations and radiation distributions are randomly generated and described in full in the appendix. The map (Figure 2a) is discretised into an 8-connected grid with side length 1.0m. The robot pose therefore comprises the known value U-MDP state features: $S_k$ is a finite set of $(x, y)$ locations $\{x, y\} \subseteq S_k$. The radiation level is a single unknown value state feature $S_e = \mathbb{R}_{\geq 0}$ where $rad\_exp \in S_e$ is the level at a location. The reactor room world is from [22] and is used with Gazebo [23] and ROS [24].

**2) Ocean currents domain**: an autonomous underwater vehicle (AUV) must navigate underwater to one of a set of two to three goal locations (multi-goal variant) or a single goal location (single-goal variant) across a 10km $\times$ 10km map, under the influence of currents. These are drawn from a real-world ocean current dataset and modelled online by a multi-output coregionalised GP [25]. The AUV is simulated by a kinematics, guidance, navigation and control (GNC) model of a small AUV. The robot pose comprises the known value U-MDP state features: $S_k$ is a finite set of $(x, y)$ locations $\{x, y\} \subseteq S_k$ on a $18 \times 20$ hexagonal grid of states, giving approximately 500m spacing between states. An example state grid with ground-truth currents is shown in Figure 2b. The unknown value state features $S_e = \mathbb{R}^2$ represent the current $x$ and $y$ velocities $\{v_x, v_y\} \in S_e$. The cost function encodes the expected traversal time between states given the AUV's water-relative velocity and the current vector values. Additional domain details and parameters are given in the appendix.

**Algorithms.** Our online BAMCP variant as described in Algorithm 1 is evaluated against two other GP belief planning algorithm baselines. The performance of sampling-based methods is dependent on the assigned compute budget to select each action, therefore we vary compute budget in each experiment. The *GP mean belief MCTS* algorithm represents a full GP belief planning approach that makes maximum likelihood assumptions. This is similar to [13], but with a GP rather than GPDM model and replacing the Monte Carlo action selection search with MCTS due to the complexity of the search problem. The *GP belief MCTS* algorithm represents the other full GP belief planning approaches [8, 3] which plan with GP beliefs inside the MCTS search tree, but sample a fixed environment dynamics instance for the MCTS rollout. This avoids carrying out belief updates during the rollout, leading to a speed up in MCTS trials per second. For all algorithms the rollout policy is to choose the action that minimises the travel distance from the next state to the closest goal state.

**Results.** The plots in Figure 3 are from multiple randomly generated problem instances: 10 for the currents domain, and 5 for the radiation domain. Each algorithm/MCTS time budget combination is given 25 repeats. Values are normalised by the expected minimum achievable cost for the corresponding randomly generated problem, calculated using an exact method and full knowledge of the environment. Due to stochasticity in the simulated robot and environment, it is possible for some runs to achieve costs below this value. Experiments are run on a 3.20GHz i7 / 64GB RAM machine.

In the ocean currents domain, Figures 3a and 3b, GP-SSP-BAMCP significantly outperforms the other algorithms. It is capable of achieving close to optimal cost with only 1 second of computation budget. Given an increased computational budget, each algorithm achieves lower cost-to-goal mean and variance. At lower computational budgets, the GP belief MCTS algorithm consistently outperforms the GP mean belief MCTS algorithm, which must carry out a belief update for each step it takes towards a goal state during a rollout. The 1000ms / GP mean belief MCTS combination is not shown, as the algorithm is not capable of building a meaningful plan within that time limit.

GP-SSP-BAMCP also significantly outperforms the baselines in the radiation domain, shown in Figures 3c and 3d. In this case, note that the increase in cost-to-goal with additional computation time is caused by the robot spending more time planning while being exposed to radiation, while the normalisation denominator is kept constant. For all methods additional computation time increases plan quality, but this can be offset by the additional stationary time spent planning. The 3000ms time budget case corresponds to the robot spending $\sim 50\%$ of its runtime planning rather than moving.

As a summary, GP-SSP-BAMCP's higher performance is due to its ability to carry out far more MCTS trials than the two full belief planning methods. On average across all sampled problem scenarios, GP-SSP-BAMCP carries out $\sim 100\times$ more MCTS trials per second than GP mean belief MCTS, and $\sim 40\times$ more than GP belief MCTS.

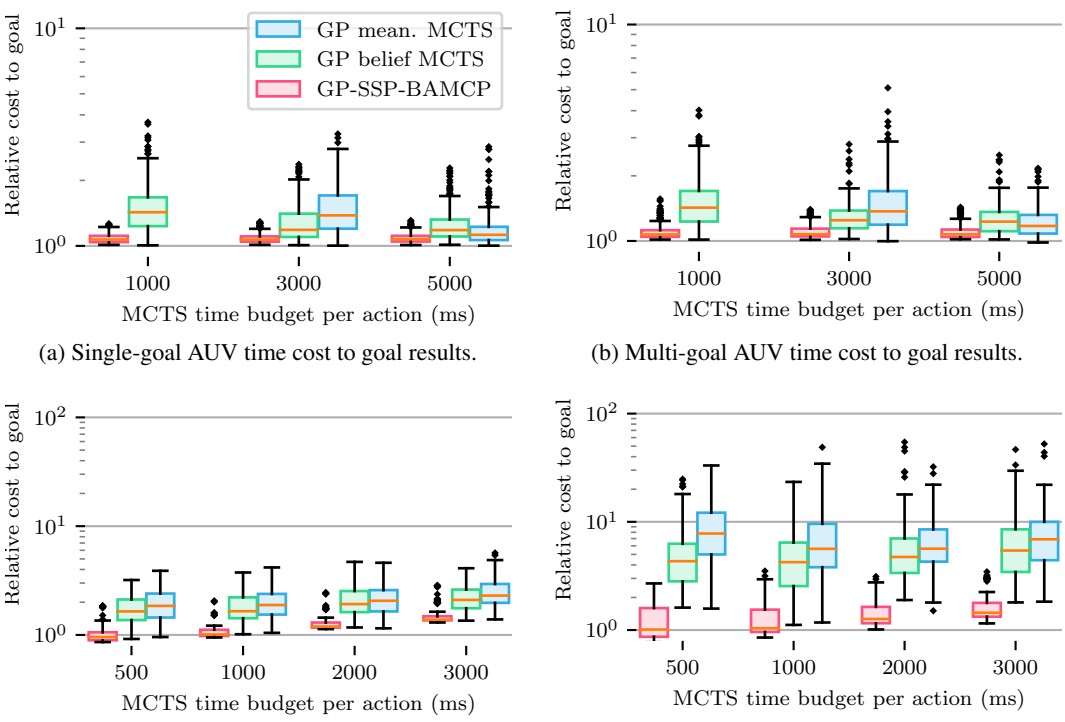

(a) Single-goal AUV time cost to goal results.

(b) Multi-goal AUV time cost to goal results.

(c) Single-goal radiation exposure cost to goal results.

(d) Multi-goal radiation exposure cost to goal results.

Figure 3: Single- and multi-goal experiment results. Plots consist of 25 simulated experiments for each algorithm/MCTS time budget combination in each randomly generated problem instance for that domain.

## 6   Conclusion

We have proposed a unified Bayesian RL framework for single-mission robot planning in GP-modelled unknown environments, and demonstrated that we are able to plan more effectively in representative real-world environments than previous approaches are able to. One potential avenue for future work is to apply progressive widening [19] or function approximation [20] techniques to our BAMCP search tree, to determine whether these produce better plans given the continuous GP-BAMDP state-space. Relaxing the assumption of negligible measurement noise would require introducing partial observability alongside transition function uncertainty. The reformulation would transform the BAMDP into a more complex Bayes-adaptive POMDP [26] with continuous GP belief. This would allow us to address settings with very high localisation uncertainty and sensor noise.

**Limitations.**   As with any method using exact GP regression, there is a limit to the size of environment that can practically be modelled. Although root sampling limits the computational cost of GP planning, exact GP regression scales with $O((n+m)^3)$ where $n$ is the number of data points and $m$ is the number of sample points. In future work we aim to replace exact GP predictions with approximate GP posterior sampling [27] to reduce the GP computational burden and improve scalability.

As our evaluation is in simulation there may exist a sim-to-real transfer gap when using the method with a real robot. This concern should be partly alleviated by our use of realistic Gazebo [23] simulation and use of real-world currents data with a complex kinematic AUV simulation. The proposed method is also a higher-level planning approach, meaning that the gap should be less wide than with low-level control methods, which are more sensitive to small sim-to-real changes.

Finally, for some environments or problem settings there is less inherent value in carrying out online planning. For example, with only a single goal state and few feasible routes to that state, the best approach for the radiation domain may be to navigate as quickly as possible to that state without incurring the radiation exposure costs of stopping to plan.

**Acknowledgments**

This work received EPSRC funding via the "From Sensing to Collaboration" programme grant [EP/V000748/1]. Matthew Budd was supported by an Amazon Web Services Lighthouse scholarship. Paul Duckworth was supported by the Cancer Research UK Radnet Oxford Centre grant (CRUK A28736).

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
