# OpenReview forum: "Bayesian Reinforcement Learning for Single-Episode Missions in Partially Unknown Environments"
_robot-learning.org/CoRL/2022/Conference — CoRL 2022 Poster_

### Official Review · Reviewer_95C2 · 2022-07-14

**Originality:** Good
**Technical Quality:** Fair
**Clarity Of Presentation:** Good
**Impact:** 2

**Recommendation:**

Weak Reject: I recommend rejecting the paper, but will not argue for my recommendation if the majority of other reviewers have a different opinion.

**Summary:**

This paper extended the existing Bayesian reinforcement learning approaches to the usage of Gaussian process belief by introducing GP model to the MDP with Unknown Feature Values. The proposed method GP-BAMCP was evaluated in two simulation tasks wiith unknown environmental features and demonstrated lower cost to goal compared with MCTS based approaches.

**Issues:**

1. the proposed method should be detailed in the form of an RL approach. The current paper mainly focused on MDP.
2. experiment should be exteneded with more analysis and additional comparsions.

**Quality Of The Limitations Section:**

Additional details required

**Reviewer Expertise:**

4: The reviewer is confident but not absolutely certain that the evaluation is correct

**Robotics Focus:**

Relevant but unlikely to deploy to hardware in near future

**Strengths And Weaknesses:**

Strengths:
This paper addressed on an interesting topic and successfully attracted the reviewer from the attached video. The structure of the paper is overall good.

Weaknesses:
The reviewer believes the approach was not detailed enough. The current paper mainly focused on introducing the MDP with GP belief but did not explained how to solve it using RL. From Algorithm 1,  the reviewer see the planning of GP-BAMCP will minimize the Q function, but there is no information about this Q function. The reviewer wish to see how the approximatied Q function and the GP model affected each other during the learning process and how the related policy and value function were built.

The experimetal result is also limited in the current version. It is difficult to know the learning capability of the proposed method without learning curve. The comparsiosn to MCTS based approaches is also not enough since both two simulation tasks are not that challenging and should be solved by many baseline RL approach even without neither Bayesian view nor GP model.  The author should clearly explain why other methods were not compared.

**Summary Of Recommendation:**

My recommendation is weak reject due to the weakness listed above.

---

> ### Author Response · Authors · 2022-08-22
> **Response to reviewer 95C2**
>
> Thank you sincerely for your review.
>
> Firstly, we would like to address your primary issue with the paper’s focus on defining the model, rather than spending most of the text describing the solution approach.
> In this work, we extend an existing Bayes adaptive model-based RL method, BAMCP [6], to solve a new model for planning missions in GP-modelled real-world environments. We therefore focus the theoretical content of the paper on providing the required evidence that our extension is valid. We specifically compare it to existing GP planning approaches, rather than re-describing the existing BAMCP algorithm. Given the limited space available to us, we believe that our decision to omit details on BAMCP in favour of focusing on describing our problem setting and providing a thorough evaluation of the methods is sensible.
>
> The BAMCP paper [6] describes the Q-function and its properties in depth: it defines the Q-function in Equation (2), and proves that the BAMCP MCTS algorithm learns the optimal Q-function of the BAMDP to ε-tolerance. We see no value repeating this content in our paper.
>
> Baseline comparison / evaluation
> ====
> We compare to the two MCTS approaches because these are the only existing approaches with a similar problem setting: *single-episode* planning in GP-modelled environments. It is not practical to compare to a ''baseline RL approach even without neither Bayesian view nor GP model”, and are not aware of any existing example of a non-Bayes-adaptive RL algorithm being applied to this problem setting.
> A standard baseline algorithm e.g. Q-learning typically assumes access to many episodes of interaction with the same environment, which is not possible in the single-episode planning setting. For example, in the radiation domain the robot must plan paths that take it outside the area it has radiation readings for.
> If one has an accurate prior model of the environment (a GP prior in our work), it is possible to sample possible environments and train an RL agent on these samples. However, as stated in Section 1, a ''standard” RL agent would have no understanding that it is encountering a different environment each episode, where the dynamics are consistent within that environment. A standard RL algorithm would learn that any feature value is possible at any location, so would converge to the expected MDP policy (i.e. always take the shortest path to the goal location). One could improve on this by training the agent on histories of observations (h, a, r, h’) of the environment features rather than just states (s, a, r, s’), but this would be equivalent to planning for all possible environments that could be encountered. This is not practical for the large domains and widely varying environment feature values encountered in our experimental domains, hence our use of online Bayes-adaptive planning.
>
> Learning curves are not informative in this online single-episode setting because the agent does not have multiple attempts to carry out the same mission in the same environment. The mission objective is the cumulative cost-to-goal, and this only has meaning at the very end of a single episode when the robot has reached a goal state.
> We therefore compare algorithms’ cost-to-goal performance across many runs given differing amounts of compute time, as is standard with single-episode learning tasks where the final cumulative cost is the performance metric, e.g. in (Guez, Arthur, David Silver, and Peter Dayan. ''Efficient Bayes-adaptive reinforcement learning using sample-based search", NeurIPS12), (Guez, Arthur, et al. ''Bayes-adaptive simulation-based search with value function approximation", NeurIPS14).
>
> We have attached the updated manuscript (including appendix) in case this is useful to you.

---

### Official Review · Reviewer_8W33 · 2022-07-26

**Originality:** Very Good
**Technical Quality:** Very Good
**Clarity Of Presentation:** Good
**Impact:** 3

**Recommendation:**

Weak Accept: I recommend accepting the paper, but will not argue for my recommendation if the majority of other reviewers have a different opinion.

**Summary:**

This paper studies a robot motion planning in a partially known environment.  A motivating application is that of an underwater robot where the location of the robot is known but the ocean currents are not.  The unknown ocean currents affect the robot dynamics.  This problem is formalized as a Markov Decision Process with a partially unknown transition function.  The authors utilize Gaussian Processes to model the belief over the unknown state.  They then adapt the Bayes-Adaptive Monte Carlo planning (BAMCP) to their problem in which there are GP beliefs.  They utilize an existing technique called “root sampling”, which avoids GP updates during rollouts.  The authors extend the technical result in [19] to show that the distribution of rollouts is, in some sense, still correct.

The results are applied to two robotics problems, a radiation domain where a robot navigates between rooms with different radiation levels, and an underwater currents domain.


**Issues:**

- In the introduction, lines 27-30, I found this discussion of RL a bit misleading as I don’t think anyone would think of training an RL agent on every possible environment.  It feels like a bit of a straw man argument.

- Lines 41-42 it is stated that BRL is known to optimally trade off exploration and exploitation.  A reference for this should be included.

- Root sampling is stated as a contribution in lines 50-51, but the concept has not yet been introduced at this point..

- In Section 4.2 the authors mention that the unknown state is considered scalar for notational simplicity.  It would be good to clarify that more than one unknown state feature is considered in the experiments.

- Line 98, the second half of the sentence appears to be repeated in the following sentence.

- Line 154, there is a discussion of implicitly making regularity and Lipschitz assumptions.  This requires some expansion as it is not clear what function must be Lipschitz continuous.

- Would it make sense to put equation (2) in a claim or lemma environment and then move the derivation to a proof?

- Line 277, typo “stoasticity”

- In the limitations there is a discussion of sim-to-real transfer.  Is imperfect localization going to be the biggest hurdle here?  This is a strong assumption and I’m unsure how it would be handled in a real implementation (say for an underwater vehicle).


**Quality Of The Limitations Section:**

Additional details required

**Reviewer Expertise:**

4: The reviewer is confident but not absolutely certain that the evaluation is correct

**Robotics Focus:**

Highly relevant to robotics but no hardware experiments

**Strengths And Weaknesses:**

Strengths:
- The simulation environment and results are quite interesting.  I appreciated that two different domains were shown.  The results compared to the two baselines are impressive.

- The main contribution is the extension of root sampling to GPs.  While this follows the development from [19], but it is a nice result that the method is applicable in this setting.

Weaknesses:
- The abstract and introduction overstate the generality of what is studied in the paper.  I recommend the introduction clearly sets the problem of a discrete state space, uncertainty only in the transition function and not cost, and with full observability of the current state.  These assumptions should be clearly laid out early in the paper.

- The notation and use of acronyms is quite heavy in places, making the paper a hard read.  I do not have a concrete suggestion for improving this as I believe much of the notation follows from [19], but it was something I struggled with in reading the paper.


**Summary Of Recommendation:**

The main contribution is to apply the results in [19] to GP beliefs over a scalar unknown quantity.  On the technical side, the extension of the results from [19] is a good contribution but does appear to follow in a straightforward manner from Lemma 1 in [19].  The simulation results are a strong point, with two interesting scenarios for testing the proposed approach.  I have some concerns about the practical applicability of the proposed approach as it requires perfect localization and that the robot remains on the gridded set of states.  However, I would like to see the paper accepted as I think the paper does have significant strengths.

---

> ### Author Response · Authors · 2022-08-22
> **Response to reviewer 8W33**
>
> We very much appreciate your review. We answer your questions & concerns in priority order below:
>
> Generality / number of unknown state features:
> ====
> Our method supports multiple unknown value state features. The underwater experiment domain has two unknown features as an example: x-direction current magnitude and y-direction current magnitude. We model these jointly with a multi-output coregionalised GP, which illustrates our point on modelling non-independent unknown feature dynamics in the first paragraph of section 4.2.
>
> Practical utility / assumed perfect localisation:
> ====
> To simplify notation, we have not included localisation uncertainty in our formulation. However, our method is able to work well in both experiment domains, which include realistic levels of localisation uncertainty. For the radiation domain this uncertainty is due to noisy Gazebo simulated LIDAR/odometry sensors and imperfect AMCL localisation, and for the underwater domain this is through simulated acoustic time-of-flight localisation with noise and variation of the speed of sound in water.
>
> This is explained by the ability to transform GP input ($\textbf{x}$) noise into output ($\textbf{y}$) noise by adding a correction term (McHutchon, Andrew, and Carl Rasmussen. "Gaussian process training with input noise", NeurIPS11). Reasonable levels of localisation uncertainty are therefore captured in our GP’s observation noise hyperparameter optimisation.
> It is true that this method would need to be adapted to tackle settings where localisation uncertainty is large compared to the characteristic lengthscales of the environment features. Combining very poor localisation with unknown environment features would result in a very difficult challenge for a mobile robot. This would require a Bayes-adaptive POMDP formulation and we have now noted this as future work.
>
> Clarity/notation:
> ====
> We are happy to address any specific place where you find the notation is overly complicated. We aim to keep the notation consistent with [19] and [4], and have otherwise kept the notation simpler by e.g. avoiding multiple unknown feature dimensions in equations (2), (3) and (4).
>
> We have attached a new version of the paper with these clarity improvements based on your feedback:
> - The introduction now clearly states the known cost function, full state observability and discrete state space assumptions,
> - Added a reference on Bayes-adaptive exploration/exploitation optimality,
> - Moved (2) and its proof to proposition/proof environments,
> - Replaced the mention of root sampling in lines 50-51 with a higher-level explanation,
> - Clarified where Lipschitz assumptions apply, and
> - Added a list of acronyms and abbreviations to the appendix to aid the reader.
>
>
> Finally, the two sentences on RL in lines 27-30 are certainly not intended to be a strawman argument. We include this because one common question from RL researchers on this work was “why is there no standard RL (e.g. Q-learning) baseline?”. We are attempting to clearly convey why “standard” non-Bayes-adaptive RL agent will have difficulty in this problem setting, e.g. by not recognising that it encounters a unique, internally consistent environment during each mission. For more discussion of this, please see our response to reviewer 95C2.

---

> > ### Comment · Reviewer_8W33 · 2022-08-25
> > **Thank you for the response!**
> >
> > Thank you for the careful response.  I missed that the underwater experiments had two unknown features.  This is an important point and addresses one of my main concerns.  I also appreciate the changes that have been made with respect to clarity and notation.  I still believe it would help the presentation if the main assumptions were collected together rather than being scattered through Sections 3 and 4. but this is a fairly minor comment.
> >
> > Overall I am positive about the paper and believe it would be a nice addition to CoRL.

---

### Official Review · Reviewer_T74h · 2022-07-31

**Originality:** Good
**Technical Quality:** Fair
**Clarity Of Presentation:** Good
**Impact:** 3

**Recommendation:**

Weak Accept: I recommend accepting the paper, but will not argue for my recommendation if the majority of other reviewers have a different opinion.

**Summary:**

The article proposed RL based planning algorithm for unknown environment with dynamics which could affect the cost of a planned trajectory. The proposed method was compared to two baseline approaches and demonstrated advantages through simulations.

**Issues:**

Could you double check on the experiments and especially on the baselines. The performance should (at least slightly) increase when more budget/planning time allowed.
Also consider adding a single target environment to enhance the comparisons.

**Quality Of The Limitations Section:**

Limitations are addressed clearly

**Reviewer Expertise:**

4: The reviewer is confident but not absolutely certain that the evaluation is correct

**Robotics Focus:**

Highly relevant to robotics but no hardware experiments

**Strengths And Weaknesses:**

Strengths:
The comparisons with two major approaches as baseline is great, which reveals the strength of the proposed approach.


Weaknesses:
The experiments part seems a bit tricky, while the total number of runs in each setup is relatively small number (10), given this is in simulation. Also the baseline approach which seems not to benefit much when more time budget is allowed. Normally if the budget increase, the performance of each method should also improve to some extent.
The multi goal/target setup is interesting, however maybe also introduce a single goal/target version to allow the audience see the full picture.




**Summary Of Recommendation:**

I understand the proposed approach may outperform the baseline according to the experiments (in simulation), however I am not sure if the baseline approach were setup correctly so even the results looks good, I am not sure if we are seeing the full picture.

---

> ### Author Response · Authors · 2022-08-22
> **Response to reviewer T74h**
>
> Thank you very much for your review. We would like to start by addressing your primary concern: the behaviour of the baselines with varying computation time in the results section.
>
> It is certainly true that MCTS algorithms should perform better on average when given a higher MCTS trial budget. This can be seen in Figure 3b for the water currents domain results, where the performance of all algorithms increases given more compute time. This trend is not seen in Figure 3a, and we explain why below.
>
> In the radiation domain, the robot incurs radiation cost while it is planning. This causes a trade-off between time spent planning (exposed to more radiation costs) and faster planning (therefore moving faster). For example, when the robot is given 3000ms MCTS time per action, it spends ~50% of its time moving and ~50% of its time planning. For all methods this additional computation improves their plan quality, but this is but this is offset by the additional stationary time.
> For our method, the minimum-cost choice of computation time seems to be ~500 or ~1000ms. For GP mean MCTS (blue), it seems to be ~2000ms.
> We have updated the explanation of this on page 7 to make this clearer.
>
> Additional points:
> ====
> We agree that the experiments would be improved by more runs per algorithm/MCTS time combination, and have increased the number of repeats from 10 to 25 per [algorithm/computation time/problem instance] combination in the new uploaded paper. This increases the total number of experimental runs shown in the paper from 1400 to 3500. The trend in the results is unchanged, with a large performance gap between our method and the baselines.
>
> Now that these additional experiments are finished, we are now running experiments for a single goal problem setup as you suggest. These results will be added to the paper before the end of the discussion period. Having only a single goal state gives less opportunity for the algorithms to demonstrate “intelligent” behaviour such as choosing the best goal, so the performance gap between methods should be smaller than with the multi-goal setup. We have added this note to the limitations section.
>
> Please find the updated manuscript (including appendix) attached to this message.

---

> > ### Author Response · Authors · 2022-08-25
> > **Updated draft with additional single-goal experiments**
> >
> > Please find attached an updated version of the paper, where we have added single-goal experiments as requested. We have also updated Section 5 to discuss the additional experiments.
> >
> > The only experiment runs remaining to be incorporated are for two single-goal radiation scenarios. The single-goal radiation environment plot (Figure 3c) is therefore temporarily composed of three, rather than five, scenarios. As soon as the remaining experiments have finished running, we will incorporate them into Figure 3c and update the draft. This will likely be close to the end of the rebuttal period, as radiation domain experiments run in real-time in Gazebo.

---

> > > ### Author Response · Authors · 2022-08-26
> > > **Updated draft with all requested experiments added**
> > >
> > > Now that all additional experiments you requested have finished running, please find the new paper draft attached to this message. This now includes all single-goal runs: 25 repeats x 5 scenarios to match the 25 repeats x 5 scenarios for the multi-goal experiments.

---

### Official Review · Reviewer_n5QS · 2022-08-02

**Originality:** Very Good
**Technical Quality:** Very Good
**Clarity Of Presentation:** Very Good
**Impact:** 3

**Recommendation:**

Strong Accept: I recommend accepting the paper and will argue for my recommendation even if other reviewers hold a different opinion.

**Summary:**

The authors propose a GP-based goal-driven BRL problem formulation for goal-driven planning in partially unknown environments. They formulate goal-driven planning in unknown environments as model-based  BRL with clear expressibility and computational efficiency.

**Issues:**

I have few questions on theoretical analysis from U-MDPs to GP-BAMDPs

- I couldn't find the reward function in the paper. Can you add them in the rebuttal?
- Can you explain how the reward is back-propagated up the tree, and the branch’s visited counters are increased?
- Is the transition function assumed to be deterministic here and independent from the objective function ?
- I was not clear on how the observation function O is sampled along trajectories? Can you elucidate it further?

**Quality Of The Limitations Section:**

Additional details required

**Reviewer Expertise:**

4: The reviewer is confident but not absolutely certain that the evaluation is correct

**Robotics Focus:**

Highly relevant to robotics but no hardware experiments

**Strengths And Weaknesses:**

Strengths:
- Paper is well written and technically sound. Motivation of the paper is solid.
- The belief update in the theoretical analysis in section 4.4 is clearly explained but I have few questions. I like the figure 1.

Weakness:
- Refer to my questions in Issues sections.

**Summary Of Recommendation:**

Technically sound paper with incremental research.

---

> ### Author Response · Authors · 2022-08-22
> **Response to reviewer n5QS**
>
> Thank you very much for your review and questions. Please find answers to your questions below.
>
> There is no reward function. We define a stochastic shortest path (SSP) MDP with a cost function C which is problem-specific. For example, in the currents domain experiment the cost is the time taken for the AUV to navigate against the (a priori unknown) water current vector at a state. Note that the cost function is assumed to be known given a value for the unknown state features, i.e., as we improve our GP estimates we also improve our cost estimates. In our problem formulation it makes sense for the cost function to be known: it would be unrealistic for e.g. the AUV to have to learn online that it slows down if it swims against current.
>
> To reduce repetition we avoided including the whole BAMCP algorithm in this paper, and refer the reader to the original (Algorithm 1, A Guez, D. Silver, and P. Dayan. "Scalable and efficient Bayes-adaptive reinforcement learning based on Monte-Carlo tree search", JAIR13). We adapt this algorithm to the SSP setting, with costs instead of rewards. The reward (or cost) propagation and visitation mechanics of BAMCP are as in standard MCTS. The visitation counters in each node are incremented when a trial’s history matches that node. Reward (or cost) is backpropagated recursively from the end of a trial. Note that the discount factor γ is 1 for SSP settings.
>
> The transition function is not assumed to be deterministic in our work. Given a single, fixed, environment, the dynamics of that environment inform our transition function and can be probabilistic if the U-MDP transition function $T^o$ is probabilistic. The U-MDP transition function $T^{o}$ and BAMDP transition function $T^{o+}$ are therefore both dependent on the environment feature function(s).
>
> Finally, note that o is not an observation function in the POMDP sense: it is an unknown function that maps state feature values in $S_k$ to state feature values in $S_e$. Sampling a possible instance of o, ô, to use in an MCTS trial is carried out by sampling from the GP posterior at the finite set of states $S_k$. This provides a value (or vector if there are multiple unknown value state features) for each unknown state feature at each combination of known state feature values. These values can then be queried at any point along the trajectory during the MCTS trial.
>
> We have attached the updated manuscript (including appendix) in case this is useful to you.

---

### Meta-Review · Area_Chair_s2vs · 2022-08-12

**Recommendation:** Accept (Poster)
**Confidence:** 5

**Metareview:**

The paper received mixed reviews. While reviewers appreciated the technical contribution as an extension of the results in [19], and the simulation results reported, they mentioned several issues that should be addressed during the rebuttal phase. In particular:

1. Clarity can be significantly improved in particular details for the Q function in Alg 1.
2. Better problem definition detailing the state space, transition function and state observability
3. Improve notation
4. Details on the baselines and adding a single target environment

================================
Post rebuttal update

The authors have adequately addressed most of the concerns by the reviewers in the revised version.

**Best Paper Nomination:**

No

---

> ### Author Response · Authors · 2022-08-28
> **Response to Meta Review and Summary of Changes**
>
> Thank you to the AC and reviewers for your feedback. As the discussion phase is now ending, here is a summary of the changes made to the paper based on this feedback. Attached to this comment is a version of the paper with the below changes highlighted in blue.
>
> For the AC's numbered issues:
>
> **1. Clarity improvements:**
>
> - *Introduction section changes*:
>   - We avoid mentioning root sampling before it has been defined (Reviewer **8W33**).
>
> - *Approach section changes*:
>   - We clarify that the Lipschitz assumptions apply to the modelled environment features (Reviewer **8W33**).
>   - (2) and (3) have been moved to proposition/proof environments as suggested (Reviewer **8W33**).
>
> - We have given an explanation of why the Q-function and other BAMCP [6]-specific components of the RL algorithm are not included in this paper in our response to Reviewer **95C2**.
>
>
> **2. Better problem definition:**
>
> - *Introduction section changes*:
>   - We have now listed all state space and observability assumptions in the introduction (Reviewer **8W33**).
> - *Preliminaries section changes*:
>   - We provide more explanation for why the cost function can be assumed to be known (Reviewer **8W33**, Reviewer **n5QS**).
> - *Limitations section changes*:
>   - We note that a Bayesian Adaptive POMDP formulation would be necessary in settings with very high localisation uncertainty, and discuss this more in our reply to Reviewer **8W33**.
>
>
> **3. Notation:**
>
> - In the Appendix we have included a new table of acronyms and abbreviations. We discuss where the notation originates in our reply to Reviewer **8W33**.
>
>
> **4. Details on the baselines and adding a single target environment:**
> - *Experiments section*:
>   - We have increased the number of experiment runs from 10 per setup to 25 per setup, and added a single-goal variant of the experiments (Reviewer **T74h**).
>   - We have drawn more attention to the online computation aspect of the radiation domain experiments sometimes giving an increase in cost given more compute time. This should address Reviewer **T74h**'s concerns on the baselines' behaviour in this domain, alongside our more detailed reply to them.